# mRNA-Loaded Lipid Nanoparticles Targeting Immune Cells in the Spleen for Use as Cancer Vaccines

**DOI:** 10.3390/ph15081017

**Published:** 2022-08-18

**Authors:** Ryoya Shimosakai, Ikramy A. Khalil, Seigo Kimura, Hideyoshi Harashima

**Affiliations:** 1Laboratory of Innovative Nanomedicine, Faculty of Pharmaceutical Sciences, Hokkaido University, Kita-12, Nishi-6, Kita-ku, Sapporo 060-0812, Japan; 2Department of Pharmaceutics, Faculty of Pharmacy, Assiut University, Assiut 71526, Egypt; 3Laboratory for Molecular Design of Pharmaceutics, Faculty of Pharmaceutical Sciences, Hokkaido University, Kita-12, Nishi-6, Kita-ku, Sapporo 060-0812, Japan

**Keywords:** mRNA, spleen, immune cell, vaccine, lipid nanoparticle

## Abstract

mRNA delivery has recently gained substantial interest for possible use in vaccines. Recently approved mRNA vaccines are administered intramuscularly where they transfect antigen-presenting cells (APCs) near the site of administration, resulting in an immune response. The spleen contains high numbers of APCs, which are located near B and T lymphocytes. Therefore, transfecting APCs in the spleen would be expected to produce a more efficient immune response, but this is a challenging task due to the different biological barriers. Success requires the development of an efficient system that can transfect different immune cells in the spleen. In this study, we report on the development of mRNA-loaded lipid nanoparticles (LNPs) targeting immune cells in the spleen with the goal of eliciting an efficient immune response against the antigen encoded in the mRNA. The developed system is composed of mRNA loaded in LNPs whose lipid composition was optimized for maximum transfection into spleen cells. Dendritic cells, macrophages and B cells in the spleen were efficiently transfected. The optimized LNPs produced efficient dose-dependent cytotoxic T lymphocyte activities that were significantly higher than that produced after local administration. The optimized LNPs encapsulating tumor-antigen encoding mRNA showed both prophylactic and therapeutic antitumor effects in mice.

## 1. Introduction

Delivering messenger RNA (mRNA) has gained substantial interest for purposes of vaccination due to its multiple advantages. In particular, it has a well-documented safety profile, since it is easily degraded in a biological environment, thus having only a transient expression and avoiding genome integration. It also has the ability to transfect non-dividing, slowly proliferating, and quiescent cells. Moreover, penetration into the nucleus, which is a rate-limiting step, is not required since mRNA translation takes place in the cytosol [1,2]. In addition, mRNA can be easily produced under good manufacturing practices. Therefore, mRNA is the genetic material of choice for immunotherapies, where the transfection of antigen-presenting-cells (APCs) is being considered.

In 2021, the world’s first mRNA drug was approved by the US FDA as a vaccine against severe acute respiratory syndrome coronavirus 2 (SARS-CoV-2). The concept of an mRNA vaccine was first described in 1993, by Martinon et al., who immunized mice with influenza antigen-encoding mRNA encapsulated in liposomes that were comprised of cholesterol (Chol), dipalmitoyl phosphatidylcholine (DPPC), and phosphatidylserine. They managed to show that, after immunization, re-stimulation of the splenocytes of the mice with virus-infected cells resulted in the development of specific cytotoxic T lymphocyte (CTL) activities. They also demonstrated this response was dependent on mRNA encapsulation and the route of administration of the mRNA-encapsulating liposomes [3]. Once APCs are transfected, they express the encoded protein, which can trigger specific activation of the immune system for the elimination of tumors or infected cells, together with an enhanced inflammation produced by the mRNA being detected as an exogenous element. However, robust, safe, and efficient delivery vectors, which are able to intracellularly deliver their cargos inside APCs, are still lacking.

The lipid nanoparticle system is one of the most promising drug delivery technologies. It is used as the platform for Patisiran, and in vaccines against SARS-CoV-2. Patisiran (ONPATTRO™) is the world’s first lipid nanoparticles siRNA product. It is a therapeutic RNA interference (RNAi) product that is used for the treatment of hereditary transthyretin (TTR)-mediated amyloidosis (hATTR) in adults [4,5]. Patisiran mainly targets liver hepatocytes through binding to endogenous lipids in the circulation and then pass the fenestrations in the blood vessels in the liver to reach hepatocytes [6]. It is composed of a pH-sensitive cationic lipid (DLin-MC3-DMA) combined with 1,2-distearoyl-sn-glycero-3-phosphocholine (DSPC), Chol, and polyethylene glycol (PEG). In this process, the LNPs target only hepatocytes and only diseases related to the liver can be treated. Thus, there is a need for developing smarter systems that, after systemic administration, have the ability to target other organs to treat diseases for which there is a great demand. Targeting the spleen is one of the most promising strategies for the development of vaccines and immunotherapies. The first step in targeting the spleen, however, is to avoid uptake by the liver and the subsequent degradation of the injected system.

LNPs used for systemic administrations of nucleic acids generally contain cholesterol, a pH-sensitive cationic lipid, phospholipids, and polyethylene glycol (PEG). Cholesterol contributes to the stability of LNPs in the blood [7,8]. Phospholipids contribute primarily to the stabilization of lipid membrane formation [9]. PEG contributes to the in vivo avoidance of phagocytic uptake and to retention in the blood [10]. The pH-sensitive cationic lipids contribute to the efficient loading of negatively charged nucleic acids during the LNP preparation by electrostatic interactions. The pH-sensitive cationic lipids also contribute to endosomal escape by membrane fusion with negatively charged endosomal membranes, which is a major barrier to the delivery of LNPs to the cytoplasm.

The currently available mRNA vaccines are injected intramuscularly to transfect muscle cells and APCs in the circulation. However, lymphoid organs, such as the lymph nodes and the spleen, contain higher numbers of APCs, which are located near B and T lymphocytes. Therefore, transfecting APCs into lymphoid organs would be expected to produce a more efficient immune response. After the systemic administration of mRNA-loaded nanosystems, most of the dose ends up in the liver. As a result, most of the particles do not reach the intended target organ, and the genetic cargo is unable to produce a therapeutic effect. A specifically designed system that targets and transfect different immune cells in lymphoid organs is needed to reach that goal. For this reason, the main goal of this work was to design mRNA-loaded LNPs that are able to efficiently transfect APCs in the spleen after intravenous (IV) administration, while simultaneously decreasing their accumulation in the liver. The success of this goal would open the possibility of developing novel genetic vaccines for immunotherapy approaches. Such a system could also be applied for the efficient and selective immune stimulation against cancer.

## 2. Results

### 2.1. Role of Helper Lipid DOPE

It was previously shown that plasmid DNA (pDNA) was successfully delivered to the spleen after IV administration using a lipid system based on the commercially available pH-sensitive lipid 1,2-dioleoyl-3-dimethylammonium propane (DODAP) [11]. DODAP, one of the first recognized pH-sensitive lipids, is not commonly used for in vivo gene delivery. Although it has a high safety profile, it is generally classified as an inefficient lipid in terms of gene expression and has no targeting ability. In this study, however, we found that DODAP could be used for efficient and targeted gene delivery in vivo, but only when combined with a specific helper lipid, dioleoylphosphatidylethanolamine (DOPE). The optimized lipid composition for pDNA spleen delivery was DODAP/DOPE/Chol/Dimyristoyl-methoxypolyethylene glycol 2000 (DMG-PEG2k) = 18.5/50/30/1.5 mol%. We prepared LNPs encapsulating mRNA using the same lipid composition as was previously optimized for pDNA. Mice were injected with the LNPs encapsulating Nanoluc (Nluc)-IRES-RFP mRNA at 0.1 mg/kg and gene expression was evaluated by measuring the luciferase activity in different organs at 24 h after injection.

The optimized composition for pDNA delivery involved the use of a combination of DODAP and DOPE. To test whether DOPE is important or not for gene expression in the case of mRNA delivery, we compared DODAP-LNPs prepared with and without DOPE. The lipid composition and characterization of the different DODAP-LNPs are shown in Table 1. DOPE-lacking LNPs (DOPE(−)) were larger in size and showed a lower encapsulation efficiency (EE) than DODAP/DOPE LNPs (DOPE(+)). DODAP LNPs prepared without DOPE caused a high gene expression in the liver and spleen while the DODAP/DOPE LNPs caused a high gene expression only in the spleen (Figure 1A). The luciferase activity in the case of the DOPE(−) LNP in the liver was higher than that for the DOPE(+) LNP. In the presence of DOPE, the gene expression was ~40-fold higher in the spleen than that in the liver. However, the gene expression for these two preparations in the spleen was not significantly different. Since the DOPE(+) LNP showed a higher spleen selective luciferase activity than the DOPE(−) LNP, we conclude that the use of DOPE in the preparation is important for achieving spleen selective gene expression. In subsequent experiments, we opted to use the DODAP/DOPE LNP as the starting lipid composition for further optimization since the objective was to find the optimum lipid composition for achieving an efficient and selective gene expression in the spleen.

### 2.2. Optimization of mRNA Delivery to the Spleen

We attempted to optimize the lipid composition for mRNA so as to improve the activity and selectivity in the spleen. The original lipid composition was similar to that optimized for delivering pDNA to the spleen, i.e., DODAP/DOPE/Chol/DMG-PEG2k, = 18.5/50/30/1.5 mol%. We examined the influence of changing the ratio of DODAP and Chol, the ratio of DOPE and DODAP, and the ratio of total lipid per amount of mRNA. The criterion for the optimal composition was the magnitude of the activity in the spleen and the extent of spleen selectivity. Different LNPs were prepared by the ethanol dilution method and the lipid film hydration method. Since the difference in the level of gene expression produced by LNPs prepared by each method was negligible (data not shown); LNPs prepared by the ethanol dilution method were used for further optimization. The characterization of different LNPs was not significantly altered by changing the lipid composition (Appendix A). We initially examined the DODAP-Chol ratio (total 48.5 mol% of total lipid) while fixing the DOPE ratio at 50 mol% (Figure 1B). The luciferase activity in the spleen was increased with increasing DODAP ratio and decreased with increasing Chol ratio, while that in the liver and lung was not affected. A higher gene expression was found in the spleen when the ratio DODAP/Chol was 48.5/0 mol%. We selected the ratio of DODAP/Chol = 38.5/10 mol% for further optimization because it is well known that cholesterol increases the stability of lipid vesicles [7].

We then checked the DOPE/DODAP ratio (total 88.5 mol%) with the cholesterol ratio being fixed at 10 mol% of the total lipid (Figure 1C). The luciferase activity in all organs was decreased with increasing DODAP and decreasing DOPE. The highest spleen activity was observed for a DOPE/DODAP ratio of 60/28.5 mol%. This result shows that, in the spleen, it is necessary to consider, not only the role of DODAP as a pH-sensitive lipid, but also that of DOPE.

We then examined the issue of the mRNA-lipid ratio (Figure 1D). The highest spleen activity was observed in the case where 240 and 320 nmol lipid were used per 10 µg of mRNA. We decided to use LNPs prepared with 240 nmol lipid in subsequent experiments to decrease the total amount of lipid used since a lower lipid to mRNA ratio is safer and more economic. The final optimized composition was DOPE/DODAP/Chol/DMG-PEG2k = 60/28.5/10/1.5 mol%, 240 nmol lipid for 10 µg of mRNA, prepared by the ethanol dilution method. LNPs prepared with the optimal lipid composition had a slightly negative charge with a particle size of about 150 nm and zeta potential of -10 mV. After the IV injection of a higher dose of mRNA (0.8 mg/kg), a high luciferase activity was found in the spleen, which exceeded 10^8^ RLU/mg protein. The activity was 41-fold higher in the spleen than that in the liver and 379-fold higher than that in the lung (Figure 1E). This indicates that the optimized LNPs can be used for efficient and selective gene expression in the spleen after IV administration.

We compared our optimized LNPs with RNA-lipoplexes (RNA-LPX); one of the strongest known systems for delivering high levels of mRNA to dendritic cells (DCs) of the spleen [12]. The RNA-LPX is formed by mixing mRNA with cationic liposomes composed of 1,2-di-O-octadecenyl-3-trimethylammonium propane (DOTMA) and DOPE (1:1) with the formation of negatively charged final particles. We confirmed the RNA-LPX was prepared successfully because the characterization of RNA-LPX was the same as reported in previous studies. The optimized DODAP-LNP or RNA-LPX was injected intravenously (0.1 mg/kg) and the luciferase activity was measured in different organs 24 h later. The RNA-LPX showed a higher gene expression in the spleen, followed by the lung, and finally the liver. The activity in the spleen was about 320-fold higher than that in the liver and about 130-fold higher than that in lung, which is considered to be a reasonable result since it has been reported that the RNA-LPX shows spleen-selective gene expression [12]. Compared to our optimized DODAP-LNP, the RNA-LPX showed about a 10 times higher luciferase activity in the spleen (Figure 1F).

### 2.3. Cellular Uptake and Gene Expression in Splenocytes

We next examined the issue of which specific cells in the spleen are internalizing the LNPs. We used specific antibodies against different types of spleen cells for this. To evaluate the cellular uptake in splenocytes, flow cytometry analyses were performed on cells isolated from the spleen. Briefly, 1 mol% DiD-labeled DODAP-LNPs were prepared as described later. These LNPs were then injected intravenously, alongside a phosphate-buffered saline (PBS) control group. After 24 hr, the mice were sacrificed, the spleens isolated, red blood cells were lysed, and the remaining cells were analyzed by flow cytometry. Flow cytometry analyses were performed on four discrete cell populations: B cells (B220+), T cells (CD3+), DCs (CD11c+), and macrophages (F4/80+) (Appendix A). In terms of the percentage of cells that had received particles, most of DCs and macrophages and approximately half of the B cells contained LNPs. On the other hand, most of the T cells failed to internalize LNPs (Figure 2A). Thus, the LNPs were mainly distributed to APCs, such as macrophages, DCs, and B cells. When comparing uptake per cell, DCs and macrophages internalized the LNPs 13- and 9-fold higher than B cells, respectively (Figure 2B). These data confirm that the LNPs are taken up by different APCs in the spleen after systemic administration.

We next evaluated the transfection activity in different cell populations inside the spleen. LNPs encapsulating mRNA-encoding Nluc were injected intravenously at a dose of 0.8 mg/kg, alongside a PBS control group. Four discrete cell populations were sorted by a cell sorter in the same manner as was used for the flow cytometry analysis. Luciferase assays were then performed on these cell populations and the luciferase activity per 3000 cells was calculated (Figure 2C). In the case of the LNP-treated group, the luciferase activity was significantly higher in DCs, macrophages, and B cells than the values for the PBS-treated group, but this was not confirmed in the case of T cells. Therefore, we successfully transfected APCs in the spleen after IV administration with mRNA without using an active targeting moiety. In T cells, B cells, macrophages, and DCs, the cellular uptake per cell, which is the geo-mean of the fluorescent dye DiD employed in LNPs in each cell, is strongly correlated with the gene expression per cell, which is the luciferase activity per 3000 cells (Appendix A).

### 2.4. Prophylactic Anti-Tumor Effect

We successfully developed LNPs that express mRNA to APCs in the spleen in vivo. We tested the potential of the developed mRNA LNPs for use as a prophylaxis against tumors. We evaluated the prophylactic anti-tumor effect of LNPs encapsulating a tumor specific mRNA. In this study, mice were treated with DODAP-LNPs encapsulating the antigen OVA-encoding mRNA (OVA-mRNA LNP), or luciferase-encoding pDNA (Luc-pDNA LNP). Control mice were treated with PBS or naked (OVA)-encoding mRNA (naked OVA-mRNA). Different LNPs and controls were injected at 1 week prior to the inoculation of tumor cells and tumor volume was then monitored (Figure 3). The OVA-mRNA LNPs completely inhibited the tumor growth in all 5 mice that were tested compared to mice that received PBS and Luc-pDNA LNP. Meanwhile, naked OVA-mRNA inhibited the tumor growth only slightly. The results reported herein confirm that the mRNA vaccine platform prepared in this study is a promising prophylactic system for preventing tumor growth. Mice body weights were not significantly changed for any group, suggesting that the administered systems have no significant toxicity (data not shown).

### 2.5. Induction of Cytotoxic T Cells

The LNP encapsulating OVA-encoding mRNA produced a strong prophylactic anti-tumor effect as shown before. To further examine the issue of whether the IV delivery of LNP induces effector cells that can specifically lyse target cells, we performed an in vivo cytotoxicity assay. At one week before the administration of target and control cells, LNPs encapsulating OVA-encoded mRNA were administered intravenously to immunize the mice, and OVA-specific cytotoxicity was then assessed by flow cytometry. The LNP encapsulating OVA-mRNA (OVA-mRNA LNP) induced strong OVA-specific cytotoxic T lymphocyte (CTL) activities and this activity was dose-dependent (Figure 4A). LNPs encapsulating firefly luciferase-encoding mRNA (Fluc-mRNA LNP) and naked OVA-encoding mRNA (naked OVA-mRNA) failed to produce a similar effect, even when a high dose (0.8 mg/kg) was used. This result indicates that OVA-mRNA LNPs are capable of inducing a high level of CTL activity even at a dose as low as 0.1 mg/kg, which is lower than the doses that were used in other published reports [13].

We also compared our developed LNPs with the RNA-LPX, which showed a high mRNA delivery to DCs of the spleen [12]. The OVA-mRNA LNPs developed in this study tended to outperform the RNA-LPX carrying the OVA-mRNA (Figure 4B), although the RNA-LPX showed about a 10 times higher luciferase activity in the spleen compared to the DODAP-LNP (Figure 1F). The difference in gene expression in the spleen appears to be not reflected by a comparable CTL activity. Compared to RNA-LPX, which mainly transfect DCs, the developed DODAP-LNPs transfected not only dendritic cells but also other APCs such as macrophages and B cells and this may activate immunity and induce a higher CTL activity.

Vaccines prepared using an LNP against novel corona viruses are administered by the intramuscular route (IM). It has been reported that the IV administration of the Bacillus Calmette–Guérin (BCG) vaccine, which is generally administered intradermally, increased its efficacy [14]. Therefore, we examined the difference in CTL activity after administration via different routes. We compared the CTL activity produced at 7 days after the IV and IM administration of the DODAP/LNP and the RNA-LPX loaded with OVA-mRNA. The results showed that both LNP and RNA-LPX tended to induce a higher CTL activity when administered intravenously (Figure 4C). The activity was higher in the case of the DODAP-LNP than in the case of RNA-LPX for both routes of administration.

### 2.6. Therapeutic Anti-Tumor Effect

We next examined the therapeutic potential of the developed mRNA vaccine against previously existing tumors. In this study, mice carrying OVA tumors were treated with DODAP-LNPs encapsulating the antigen OVA-encoding mRNA (OVA-mRNA LNP), or luciferase-encoding mRNA (Luc-mRNA LNP) and the RNA-LPX loaded with OVA-encoding mRNA (RNA-LPX). Control mice were treated with PBS or the naked (OVA)-encoding mRNA (naked OVA-mRNA). Different nanoparticles and controls were injected at 8, 11, and 14 days after the inoculation of tumor cells and tumor volume was then monitored (Figure 5). The results showed that the OVA-mRNA LNPs inhibited tumor growth more strongly than the other groups, which confirms the therapeutic effect of the developed LNPs. The higher antitumor effect obtained with the developed DODAP-LNP system compared to that obtained with the efficient RNA-LPX system confirms the superiority of using LNPs encapsulating mRNA compared to using lipoplexes prepared by simply mixing mRNA with cationic liposomes.

## 3. Discussion

Developing mRNA vaccines is of great interest and is currently receiving considerable attention for prophylaxis against infectious diseases as well as against tumors. New efficient mRNA-based COVID 19 vaccines are now being approved and are currently being used as the first line of defense against COVID infections. These mRNA vaccines are based on lipid nanoparticles which are injected intramuscularly where they transfect myocytes to produce the spike protein of the virus. They can also transfect APCs that are located near the injection site resulting in antigen presentation through major histocompatibility complex (MHC) proteins which activate antibody production as well as the production of CD8+ T lymphocytes. The transfection of APCs in the spleen or lymph nodes is expected to result in more efficient antigen production, thus leading to a stronger prophylactic effect. However, targeting immune cells in the spleen or lymph nodes is not an easy task and novel gene delivery systems need to be developed for this purpose. The aim of this study was to develop lipid-based nanoparticles encapsulating mRNA which have the ability to transfect immune cells in the spleen after IV administration.

We prepared mRNA-loaded lipid nanoparticles in which the main lipid component was the pH-sensitive lipid DODAP and we investigated their transfection activities in different organs after IV administration. We investigated the effect of the inclusion of the helper lipid DOPE on transfection activities in different organs. Since DOPE is a fusogenic lipid, this property contributes to the overall transfection activity by enhancing the endosomal escape [15]. DOPE-containing LNPs showed higher spleen selective transfection activity than LNPs lacking DOPE (Figure 1). The presence of DOPE caused a significant decrease in the gene expression in the liver, which enhances the spleen selectivity. It has been reported that LNPs formulated with DOPE interact strongly with apolipoprotein E (ApoE) and accumulate in increased amounts in the livers of injected mice compared to LNPs formulated with DSPC, which interact more weakly with ApoE and accumulated to a greater degree in the spleens of the injected mice [16]. Therefore, it is thought that the inclusion of DOPE would decrease the activity in the liver, but activity was not decreased in the spleen. The exact reason for this is unclear, but it is possible that the structure of the LNPs, the protein corona on the LNPs, or the intracellular dynamics are linked to the spleen selectivity of DOPE.

On the other hand, DODAP is one of the earliest used pH-sensitive cationic lipids used for nucleic acid encapsulation, but the gene delivery efficiency is known to be low [17,18]. The potency of an LNP encapsulating a siRNA system was improved by replacing DODAP with DLinMC3DMA; a more efficient pH-sensitive lipid [19,20,21]. The low activity of DODAP-LNPs in vitro is mainly attributed to its low level of cellular internalization and low endosomal escape, compared to conventional cationic liposomes as well as other recently synthesized pH-sensitive cationic lipids. Surprisingly, DODAP-LNPs encapsulating siRNA do not produce a strong gene silencing effect in the spleen after systemic administration, but DODAP-LNPs encapsulating mRNA show a selective gene expression in the spleen, as shown in this study. DODAP-LNPs encapsulating pDNA also showed enhanced spleen selective gene expression [11].

We performed an optimization study of the lipid composition of mRNA-loaded LNPs in an attempt to obtain the highest gene expression in the spleen. After an examination of the effect of the ratio of DODAP/Chol, we selected a DODAP/Chol ratio of 38.5/10 instead of 48.5/0 because cholesterol is a commonly used ingredient in liposomes and LNPs as a component that stabilizes lipid bilayer by filling in the gaps between phospholipid molecules [7]. Using cholesterol as 30 mol% of the total lipid resulted in the most stable formulation to guarantee the controlled and reproducible release of drugs; however, increasing the cholesterol ratio decreased transfection activities in the spleen (Figure 1B).

After optimization, we evaluated the cellular uptake and the luciferase activity in splenocytes. It is well known that macrophages and dendritic cells internalize LNPs by phagocytosis. However, gene expression efficiency is not high in these cells [12,22]. On the other hand, lymphocytes (T and B cells) do not internalize LNPs efficiently and the transfection of these cells in vivo is difficult [23]. The findings reported in this study show that the LNPs were internalized by most of the macrophages and DCs and a significant fraction of B lymphocytes also internalized the LNPs. In the first study of mRNA LNPs that were capable of inducing protein expression within B lymphocytes in vivo, about 7% of the total B cell population was labeled with the Cy5 mRNA at one hour after a single intravenous injection [24]. In our study, however, we measured cellular uptake after 24 h, and about 50% of the total B cell population had internalized the LNPs. LNPs were not internalized by T lymphocytes, probably due to the limitation of blood supply to T cells in the spleen.

For purposes of vaccination, the main goal is the transfection of APCs, such as dendritic cells, macrophages, and B cells. These cells will trigger a specific immune response against encoded antigens, thus producing an immune attack against cells that express these antigens. This ends up with their elimination and the reduction in tumor size, in the case of cancer, for example [25]. Although the cellular uptake of LNPs in the case of DCs was lower than that for macrophages in terms of fluorescence measured per cell, the luciferase activity produced in macrophages and DCs was similar (Figure 2).

Only a few reports have appeared that confirm the transfection of mRNA in B cells after IV injection, but the transfection efficiency was low [24,25]. In this study, B cells were observed to be efficiently transfected with mRNA. The transfection of B cells may contribute to the increased immune response. It is also possible that types of cells, other than immune cells, may be transfected. In the case of nanoparticles capable of delivering mRNA to the spleen, gene expression has been confirmed in cells other than immune cells, including endothelial cells [26]. However, transfection of these cells would not be expected to affect the immune response.

In the prophylactic anti-tumor effect, naked OVA-mRNA inhibited tumor growth to a slight extent (Figure 3). The mRNA molecule is recognized by the toll like receptor 7/8 (TLR7/8) and induces a type I interferon response, which has an immunostimulatory effect. Therefore, it is generally thought that the naked mRNA group may activate the innate immune system and suppress tumor growth. On the other hand, the optimized mRNA loaded LNPs induced strong antitumor activity. This may be due to the roles of both the expression of the target protein and the adjuvant effects in antigen-presenting cells, which induced a strong OVA antigen-specific immune response. The optimized OVA-mRNA LNP was found to induce a high CTL activity, even at a dose as low as 0.1 mg/kg (Figure 4). Concerning the prophylactic anti-tumor effect, this LNP would be expected to be effective at even lower doses. In addition, since naked-mRNA could not induce an OVA-specific CTL, this supports the conclusion that naked-mRNA failed to cause an antigen-specific immune response in the prophylactic anti-tumor effect.

Although the transfection activity in the spleen for the RNA-LPX was higher than that produced from the optimized DODAP-LNP, the CTL activity produced from the RNA-LPX and the DODAP-LNP, at a high dose, was comparable (Figure 1 and Figure 4). It should also be noted that the activity obtained for a low dose of the mRNA-LNP was higher than that produced from the mRNA-LPX. The superiority of DODAP-LNP at low dose was further confirmed when comparing the CTL activity produced after IV or IM administration. Screening the efficiency using gene expression in the entire spleen as an indicator may not directly lead to therapeutic effects. The RNA-LPX may transfect cells other than immune cells in the spleen, which would increase the gene expression in the spleen. In addition, the findings show that the RNA-LPX mainly transfected DCs in the spleen [12], while DODAP-LNP transfected not only DCs, but also macrophages and B cells. The transfection of B cells may have affected the immune response, as indicated by the screening of the CTL activity. In the case of the development of nanoparticles targeting the spleen, screening using gene expression in the spleen may be useful; however, screening using CTL as an indicator of the immune response is more desirable in regard to the development of mRNA vaccines. Measuring gene expression in immune cells may also be used as an indicator of the efficiency of delivery. In T cells, B cells, macrophages, and DCs, the geo-mean for the fluorescent dye DiD employed in the LNPs in each cell was found to be strongly correlated with the luciferase activity per 3000 cells. Therefore, by assessing the cellular uptake of each cell, it is possible to predict the gene expression of each cell type.

In the therapeutic anti-tumor experiment (Figure 5), the DODAP-LNP produced a stronger antitumor effect compared to the RNA-LPX. In a previous report, the RNA-LPX was shown to have therapeutic effects in B16F10 lung metastasis models, but the dose used was approximately six times higher than the dose used in the study (40 µg/mouse) [12]. Therefore, the DODAP-LNP that was developed and evaluated in this study proved to be more superior to RNA-LPX in terms of CTL activity and antitumor effect.

Both LNP and RNA-LPX tend to induce a higher CTL activity when administered intravenously than that when administrated intramuscularly (Figure 4). It has been reported that the IV administration of RNA-LPX induced a higher number of OVA-specific CD8+ T cells compared to subcutaneous administration. This may be due to the higher number of immune cells that are activated by targeting the spleen rather than the circulating APCs or the inguinal lymph nodes when comparing local and systemic administration.

## 4. Materials and Methods

### 4.1. Materials

DOPE, DOTMA, and DMG-PEG were purchased from the NOF CORPORATION (Tokyo, Japan). DODAP and Chol were purchased from Avanti Polar Lipids (Alabaster, AL, USA). Ribogreen assay system was purchased from Thermo Fisher Scientific. Phycoerythrin (PE)-labeled anti-mouse CD3 antibody, PE-labeled anti-mouse CD11c antibody, fluorescein isothiocyanate (FITC)-labeled anti-mouse CD45R/B220 antibody, FITC-labeled anti-mouse F4/80 antibody, purified anti-mouse CD16/32 were purchased from Biolegend (San Diego, CA, USA). 1,1′-dioctadecyl-3,3,3′,3′-tetramethylindodicarbocyanine (DiD) was purchased from Invitrogen (Carlsbad, CA, USA). The luciferase assay reagent and reporter lysis buffer were obtained from Promega (Madison, WI, USA). Dulbecco’s modified eagle medium (DMEM), Roswell Park Memorial Institute medium (RPMI), and fetal bovine serum (FBS) were purchased from Sigma-Aldrich Co. (St. Louis, MO, USA). All other materials were of reagent-grade and were commercially available.

### 4.2. Animals

Four-week-old male ICR mice or female C57BL/6J mice were purchased from Sankyo Labo service (Shizuoka, Japan). The experimental protocols used in this study were reviewed and approved by the Hokkaido University Animal Care Committee in accordance with the “Guide for the Care and Use of Laboratory Animals”. In all experiments, the animals were used without fasting. Mice were euthanized by cervical dislocation.

### 4.3. Preparation of LNP

The LNPs used in this study were prepared by the ethanol dilution method. An amount of 10 µg mRNA was dissolved in 250 µL 10 mM HEPES buffer (pH 4.0). The lipids were dissolved in ethanol (total of 250 µL). While stirring the lipid ethanol solution, 250 µL 10 mM HEPES buffer (pH 4.0) containing 10 µg mRNA, 500 µL 10 mM HEPES buffer (pH 4.0) and 1 mL 10 mM HEPES buffer (pH 4.0) × 4 were added in that order. The LNPs solution was centrifuged (25 °C, 1800× *g*, 30 min) with an Amicon tube ultra 100k. LNPs were recovered from the tubes with PBS.

### 4.4. Characterization of LNPs

The average size (nm) and zeta-potential of the LNPs were determined using a Zetasizer Nano (Malvern, UK). The mRNA encapsulation efficiency and recovery were determined by a Ribogreen assay, LNPs were diluted in 10 mM HEPES buffer pH 7.4 containing 1 mg/mL dextran sulfate and Ribogreen reagent in the presence or absence of 10 w/v% Triton X-100. Fluorescence intensity was measured by INFINITE 200 instrument with λem = 480 nm, λex = 515 nm. mRNA concentrations were calculated from mRNA standard curves. mRNA encapsulation efficiency was calculated by comparing the mRNA concentration in the presence and absence of Triton X-100.

### 4.5. Preparation of RNA-LPX

DOTMA/DOPE (1:1 mol ratio, total 400 µL) dissolved in ethanol to a total lipid concentration of 10 mM was added to a glass test tubes and the solvent was removed by an evaporator. After adding 800 µL of D-PBS (-), the contents of the tube were hydrated for about 2 min. The mixture was then agitated and sonicated in a bath sonicator for about 5 min. The prepared liposome solution was diluted 5-fold with D-PBS (-). A 0.173 mg mRNA/mL (28.875 µL) of mRNA solution diluted in D-PBS (-) was added to liposome solution (21.125 µL) with stirring (nitrogen/phosphate (N/P) = 1.3/2). The characterization of RNA-LPX was measured in the same way as LNPs.

### 4.6. Gene Expression In Vivo

Four-to-five-week-old ICR mice were treated at a dose of 0.1 or 0.8 mRNA mg/kg body weight (concentration was 26.6 ± 5.6 mRNA µg/mL). The mice were sacrificed at 24 h after the treatment, and the liver, lungs, and spleen were collected. The collected organs were washed with saline, weighed, and minced with scissors. A 0.2 g sample of liver tissue and the other whole collected organs were completely homogenized using a MicroSmash homogenizer in 1 mL of lysis Buffer (100 mM Tris-HCl, 2 mM EDTA, 0.1% Triton X-100, pH 7.8). After centrifugation at 15,000 rpm for 10 min at 4 °C, a 25 µL aliquot of the supernatant was examined for luciferase activity. The amount of protein in the samples was determined using a BCA protein assay kit (Pierce). Luciferase activities are expressed as relative light units (RLU) per mg of protein.

### 4.7. Gene Expression by Cell Type in the Spleen

Four-to-five-week-old ICR mice were treated with a dose of 0.8 mRNA mg/kg body weight. The mice were sacrificed at 24 h after the treatment, and the liver, lung, and spleen were collected. The collected organs were then washed with saline, weighed, and minced with scissors. Splenocytes were collected and centrifuged at 500× *g* for 5 min at 4 °C. After removing the supernatant, 1 mL ACK lysis was added and the preparation incubated for 5 min at room temperature. A volume of 9 mL of RPMI medium for the spleen (5 mL P/S, 5 mL 1 M HEPES buffer solution, 5 mL 1 mM sodium pyruvate solution, 500 µL 2-Mercaptehanol in 500 mL RPMI-1640 low glucose medium) was added over a period of 5 min at 4 °C. After removing the supernatant, 10 mL RPMI medium was added and washed twice. After cell counting, 1 × 10^6^ cells were blocked with 1.0 µL CD16/32 antibodies in 50 µL FACS buffer and incubated 10 min in ice. The cells were then stained with antibodies for more than 30 min on ice. After incubation, cells were washed with FACS buffer and then cells were sorted via flow cytometry. Luciferase activity was then analyzed.

### 4.8. E.G-7 OVA Cell Cultures 

For the E.G-7 OVA cell culture, RPMI 1640 medium containing 50 µM 2-mercaptoethnol, 10 mM HEPES, 1 mM sodium pyruvate, 100 U/mL penicillin-streptomycin, and 10% FBS was used. To start the culture, the cell suspension was quickly lysed at 37 °C, diluted with medium (9 mL), and then centrifuged at 500× *g*, 3 min. The supernatant was removed, the cells were resuspended in medium (10 mL), seeded in 10 cm dishes, and cultured under 37 °C in an atmosphere of 5% CO_2_ conditions. Passaging was performed every two days. For passaging, cells were collected and then centrifuged at 500× *g* for 3 min. After the supernatant was removed, the cells were re-suspended in medium and counted. For counting, the cell suspension was mixed with an equal volume of 0.5% trypan blue solution, and only the unstained viable cells were counted.

### 4.9. Anti-Tumor Prophylactic and Therapeutic Experiments

The mRNA encoding antigen (ovalbumin; OVA) was purchased from Trilink (San Diego, CA, USA). In the prophylactic anti-tumor effect experiment, all groups were administered to the C57BL/6 mice by intravenous injection for immunization at 1 week before the tumor inoculation. In the therapeutic anti-tumor effect experiment, all groups were administered to the C57BL/6 mice by intravenous injection at 8, 11, and 14 days after the tumor inoculation. For tumor inoculation, 5 × 10^5^ antigen (OVA)-expressing E.G7-OVA cells were subcutaneously injected. Tumor volume was calculated by the following formula: (major axis × minor axis2) × 0.52.

### 4.10. Antigen-Specific Cytotoxic Effect

Antigen specific cytotoxic effect was measured as previously described [11]. C57BL6J-female-4-5-week-old mice were intravenously administered with PBS, naked OVA-encoding mRNA, OVA-encoding mRNA prepared as lipolexes, or the DODAP-LNPs encapsulating mRNA-encoding firefly-luciferase (on day 0, dose: 0.8 mg/kg) or DODAP-LNPs encapsulating OVA-encoding mRNA (on day 0, dose: 0.05, 0.075, 0.1, 0.8 mg/kg). At 5 days after the administration, the naïve C57BL6J mice were sacrificed and splenocytes were collected. Half of the splenocytes were treated with the OVA257-264 peptides (1 mM) for 1 h at 37 °C. The untreated and peptide-treated cells were labeled with 0.5 µM or 5 µM carboxyfluorescein succinimidyl ester (CFSE) (Invitrogen), respectively, in PBS for 10 min at 37 °C. Equal numbers (5 × 10^6^) of CFSE-low (untreated) and CFSE-high (peptide-treated) cells were mixed and intravenously injected into the immunized mice. Splenocytes from the immunized mice were collected 20 h after injection, and single-cell suspensions were analyzed by flow cytometry (CytoFLEX). The numbers of CFSE-low and CFSE-high cells were used to calculate the percentage of peptide-treated cell killing. Peptide-treated cell specific killing was determined as the percentage of specific lysis = (1−control ratio/experimental ratio) × 100.

## 5. Conclusions

In this study, the goal was to develop LNPs whose activity is selective for the spleen. We succeeded in the selective transfection of mRNA to the spleen after IV administration. The optimized composition of the LNP was found to be DOPE/DODAP/Chol/PEG-DMG2k = 60/28.5/10/1.5 mol%, 240 nmol lipid for 10 µg mRNA. Dendritic cells, macrophages, and B cells in the spleen were all successfully transfected. The developed DODAP-LNPs produced a strong prophylactic effect as well as a therapeutic antitumor effect. Although a lower gene expression was produced in the spleen compared to RNA-LPX, the DODAP-LNPs produced a higher CTL activity and higher antitumor effect. Therefore, we conclude that the developed LNPs are highly promising for use in the treatment of cancer, infections, and other diseases.

## Figures and Tables

**Figure 1 pharmaceuticals-15-01017-f001:**
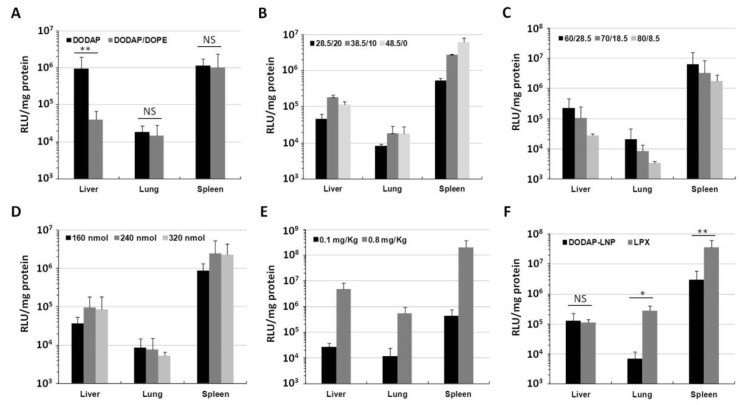
Luciferase activity in vivo. (**A**) Luciferase activity of DODAP-LNP prepared with or without DOPE. The LNP that was injected contained 0.1 mRNA mg/kg and luciferase activity was measured 24 h after injection. Luciferase activity is expressed as relative light unit (RLU) per mg of total protein (** *p* < 0.01, NS: not significant, two-tailed unpaired *t* test). (**B**–**D**) Optimization of the lipid composition in vivo. (**B**) The ratio of DODAP and cholesterol (DOPE and DMG-PEG were fixed at 50 and 1.5 mol% of total lipid, respectively. (**C**) The ratio of DOPE and DODAP (Chol and DMG-PEG were fixed at 10 and 1.5 mo% of total lipid, respectively). (**D**) The ratio of the amount of total lipid to 10 µg mRNA. (**E**) The luciferase activity measured 24 h after injection of 0.1 and 0.8 mg/kg. (**F**) The luciferase activity 24 h after injection of DODAP-LNP and RNA-LPX loading Nluc-IRES-RFP mRNA (** *p* < 0.01, * *p* < 0.05, NS: not significant, two-tailed unpaired Student’s *t*-test). Each bar represents the mean +/− SD of at least 3 different experiments).

**Figure 2 pharmaceuticals-15-01017-f002:**
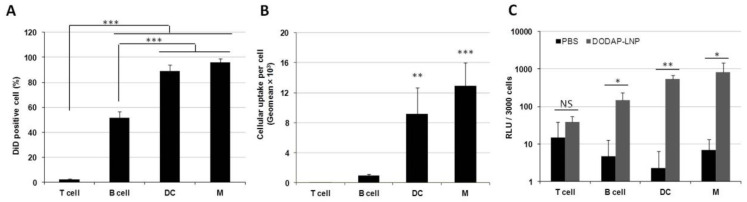
The cellular uptake and gene expression in splenocytes. Different LNPs were prepared by the ethanol dilution method and labeled with 1 mol% DiD. Splenocytes were isolated at 24 h after the injection of 400 µL LNPs. (**A**) DiD positive cells (%) in each cell type, *** *p* < 0.001, Tukey-Kramer test. (**B**) Geo-mean fluorescence intensity in each cell type, *** *p* < 0.001, ** *p* < 0.01, Tukey-Kramer test. (**C**) Luciferase activity per 3000 cells, * *p* < 0.05, ** *p* < 0.01, NS: not significant, Student *t* test.

**Figure 3 pharmaceuticals-15-01017-f003:**
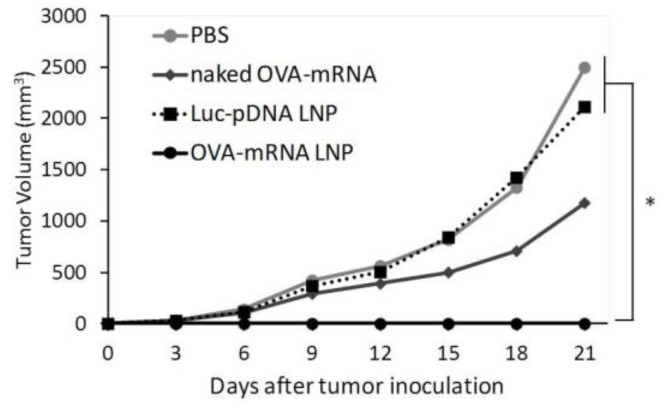
Prophylactic anti-tumor effect. C57BL/6J mice were treated with PBS, naked OVA-encoding mRNA (naked OVA-mRNA), LNP encapsulating luciferase-encoding pDNA (Luc-pDNA LNP), and LNP encapsulating OVA-encoding mRNA (OVA-mRNA LNP) at 7 days before tumor inoculation. The mice were inoculated with E.G7-OVA cells and tumor volume was monitored. The plots represent the mean +/− SEM (total of 5 mice/group, * *p* < 0.05, Tukey-Kramer test).

**Figure 4 pharmaceuticals-15-01017-f004:**
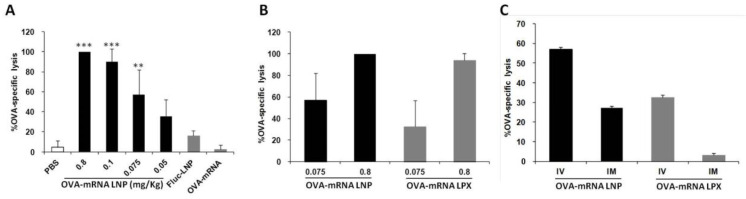
CTL analysis after administration of different doses of mRNA formulations using different routes of administration. (**A**) OVA-specific CTL activity measured after treatment with different doses of OVA-mRNA LNP. LNP encapsulating luciferase gene (Fluc-LNP) or naked OVA mRNA were used as controls at a dose of 0.8 mg/kg (*** *p* < 0.001, ** *p* < 0.01, vs. PBS, Fluc-mRNA LNP and naked OVA-mRNA, Tukey-Kramer). (**B**) Comparison of OVA-specific CTL activity between OVA-mRNA LNP and RNA-LPX using different dose. (**C**) Comparison of OVA-specific CTL activity between IV and IM injection of OVA-mRNA LNP and RNA-LPX. The administered dose was 0.075 mg/kg.

**Figure 5 pharmaceuticals-15-01017-f005:**
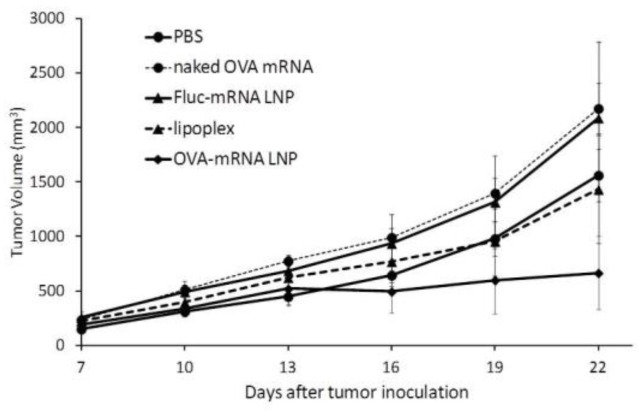
Therapeutic anti-tumor effect. C57BL/6J mice were inoculated with E.G7-OVA cells. The mice were treated with different solutions at 8, 11, and 14 days after tumor inoculation and tumor volume was monitored. Mice groups were treated with PBS, naked OVA-encoding mRNA (naked OVA-mRNA), LNP encapsulating luciferase-encoding mRNA (Luc-mRNA LNP), LNP encapsulating OVA-encoding mRNA (OVA-mRNA LNP), and RNA-LPX loading OVA-encoding mRNA (RNA-LPX). The plots represent the mean +/− SEM (total of 5 mice/group, NS: not significant, Tukey-Kramer test).

**Table 1 pharmaceuticals-15-01017-t001:** Characterization of DODAP-LNPs prepared with or without DOPE ^1^.

LNP ^2^	Size (nm)	PDI	ζ-Potential (mV)	EE (%)
DODAP	186 ± 16	0.240 ± 0.083	−9.8 ± 17	49.4 ± 7.4
DODAP/DOPE	125 ± 22	0.013 ± 0.045	−10 ± 8.6	83.3 ± 13

^1^ Values are mean ± SD of at least three different preparations. ^2^ Total lipid; 240 nmol for 10 µg mRNA, Preparation method; ethanol dilution method.

## Data Availability

Data is contained within the article and Appendix A.

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
