# Peer review of "mRNA-Loaded Lipid Nanoparticles Targeting Immune Cells in the Spleen for Use as Cancer Vaccines"

_pharmaceuticals, 2022, doi:10.3390/ph15081017_

Round 1

Reviewer 1 Report

Authors proposed a paper entitled “mRNA-loaded lipid nanoparticles targeting immune cells in the spleen for use as cancer vaccines” for the publication in Pharmaceuticals, mdpi.

The paper has a good scientific soundness.

The use of English is quite good

I suggest the addition of an abbreviation list, according to the guidelines of this Journal.

I only have some minor issues

Line 71. “It is used as the platform for Patisiran”. Is this currently used in this formulation?

Line 75. “in vivo” should be written in italique.

Line 100. “In our laboratory”. I would suggest to use more impersonal forms.

Line 123. “40-fold higher in the spleen” were there any supporting references?

Table 1. Was it possible to investigate the effect of different DODAP/DOPE ratio?

Line 218. Please, define PBS.

Line 282. “in vivo” should be written in italique.

Figure 4 caption could be enlarged with expanded and more specific concepts.

Figure 5. Improve the focus of this diagram.

Line 505. Please add manufacturer name and country for this instrument.

Line 617. “Author 1, A.B.; Author 2, C.D. Title of the article. Abbreviated Journal Name Year, Volume, page range.” This was included in author guidelines and should be removed.

Perhaps Authors should add declarations about the use of animals for research purposes; please, check the guidelines for this journal.

Author Response

Authors proposed a paper entitled “mRNA-loaded lipid nanoparticles targeting immune cells in the spleen for use as cancer vaccines” for the publication in Pharmaceuticals, mdpi.

The paper has a good scientific soundness.

The use of English is quite good.

We thank the reviewer for his/her positive evaluation of our manuscript.

I suggest the addition of an abbreviation list, according to the guidelines of this Journal.

Based on the reviewer suggestion, we added a list of abbreviations in the revised version after the references section (Line 693).  

I only have some minor issues

Line 71. “It is used as the platform for Patisiran”. Is this currently used in this formulation?

Yes.

Patisiran is a lipid nanoparticles system encapsulating siRNA for delivery to hepatocytes. In this regard, it is the same platform used in this study, which encapsulates mRNA. The main difference in lipid composition is the type of pH-sensitive lipid used. Patisiran uses DLin-MC3-DMA while we use DODAP. Using DODAP is the main reason of spleen targeting since DLin-MC3-DMA lipid is known to cause hepatocytes targeting. Patisiran also contains DSPC, cholesterol and PEG2K. Our optimized system contains DOPE, cholesterol and PEG2K.

We added the composition of Patsiran in the revised version (Lines 61-64)

Line 75. “in vivo” should be written in italique.

We modified the word “in vivo” throughout the manuscript.

Line 100. “In our laboratory”. I would suggest to use more impersonal forms.

We changed the expression to (It was previously shown that …) (Line 102).

Line 123. “40-fold higher in the spleen” were there any supporting references?

This is an explanation of the result which appears in Figure 1A. We believe no other reference compared the gene expression of DODAP LNPs (+/- DOPE) in different organs after IV administration.

Table 1. Was it possible to investigate the effect of different DODAP/DOPE ratio?

Figure 1C showed the effect of different DODAP/DOPE ratios on gene expression. The luciferase activity in all organs was decreased with increasing DODAP and decreasing DOPE. The highest spleen activity was observed for a DOPE/ DODAP ratio of 60/28.5 mol%. Characterization of LNPs prepared with different DODAP/DOPE ratios appear in Supplementary table 1.

Line 218. Please, define PBS.

We defined PBS in the revised manuscript (Line 212).

Line 282. “in vivo” should be written in italique.

We modified the word “in vivo” throughout the manuscript.

Figure 4 caption could be enlarged with expanded and more specific concepts.

We changed the caption based on the reviewer suggestion to: CTL analysis after administration of different doses of mRNA formulations using different routes of administration. (Line 314)

Figure 5. Improve the focus of this diagram.

We changed Figure 5 to use more clear points and lines.

Line 505. Please add manufacturer name and country for this instrument.

We added the manufacturer name and country (Malvern, USA). (Line 514)

Line 617. “Author 1, A.B.; Author 2, C.D. Title of the article. Abbreviated Journal Name Year, Volume, page range.” This was included in author guidelines and should be removed.

We removed this sentence.

Perhaps Authors should add declarations about the use of animals for research purposes; please, check the guidelines for this journal.

The original submission contains a declaration about the animal use for research purposes (The experimental protocols used in this study were reviewed and approved by the Hokkaido University Animal Care Committee in accordance with the “Guide for the Care and Use of Laboratory Animals”.) (Lines 498-500)

Reviewer 2 Report

Dear Author,

The manuscript entitled “mRNA-loaded lipid nanoparticles targeting immune cells in 2 the spleen for use as cancer vaccines” is well written and presents interesting findings. Please address the following comments:

Comment 1:

Introduction: The Author started the introduction with a Patsiran discussion (lien 31 to 41); I would suggest starting the introduction with more relevant information like the cancer vaccine and its limitation in terms of delivery.

Comment 2:

What was the final concentration of mRNA in the final formulation? 

Comment 3:

Section 2.1 Role of helper lipid DOPE:  The Author stated in this section that the gene expression for these two preparations (DOPE(+) LNP and DOPE(-) LNP) in the spleen was not significantly different. It means that each formulation has the same effect on the spleen immune response. However, including the DOPE in formulation reduced the immune response in the liver. If I am correctly understanding, the inclusion of DOPE reduces the overall response in the liver but has the same effect on the spleen. Then how did the preparations (DOPE(+) LNP superior compared to preparations (DOPE(-) LNP?

Comment 4:

How did the Author choose 0.1 and 0.8 mg/Kg dose mRNA for studies?

Comment 5:

How did DODAP selectively accumulate the mRNA in the spleen? What is the mechanism behind it?

Author Response

Dear Author,

The manuscript entitled “mRNA-loaded lipid nanoparticles targeting immune cells in 2 the spleen for use as cancer vaccines” is well written and presents interesting findings. Please address the following comments:

We thank the reviewer for his/her positive evaluation of our manuscript.

Comment 1:

Introduction: The Author started the introduction with a Patsiran discussion (lien 31 to 41); I would suggest starting the introduction with more relevant information like the cancer vaccine and its limitation in terms of delivery.

Based on the reviewer suggestion, we changed the introduction to start with the mRNA use for vaccination. The discussion about Patisiran comes next when we introduce LNPs as one of the promising gene delivery systems.

Comment 2:

What was the final concentration of mRNA in the final formulation?

It was 26.6 ± 5.6 mRNA µg/mL (We calculated the volume injected after considering the EE and recovery ratio). We added this information to the revised manuscript (line 534).

Comment 3:

Section 2.1 Role of helper lipid DOPE:  The Author stated in this section that the gene expression for these two preparations (DOPE(+) LNP and DOPE(-) LNP) in the spleen was not significantly different. It means that each formulation has the same effect on the spleen immune response. However, including the DOPE in formulation reduced the immune response in the liver. If I am correctly understanding, the inclusion of DOPE reduces the overall response in the liver but has the same effect on the spleen. Then how did the preparations (DOPE(+) LNP superior compared to preparations (DOPE(-) LNP?

It is superior in terms of reducing off-target responses/side effects. When we target the spleen, it is important to avoid transfecting liver cells, which may produce live-related toxicity such as increased liver enzymes. The DODAP/DOPE-LNPs selectively target genes into antigen-presenting cells in the spleen (Figure 2C), with low level of liver expression (Figure 1). The liver composes of mainly hepatocytes (~80%), and the high liver expression of DOPE(-)-LNP may be produced from the hepatocytes. Gene expression in cells other than immune cells, the target of this study, can lead to toxicity including inflammatory reactions. Taken together, DOPE(+)-LNP may be superior compared to DOPE(-)-LNP with respect to the low level of liver toxicity.

Comment 4:

How did the Author choose 0.1 and 0.8 mg/Kg dose mRNA for studies?

0.1 mg/kg was the dose used during the lipid composition optimization process. We selected to use the lowest possible dose in optimization to reduce the cost and possible side effects to mice. 0.8 mg/kg is the maximum dose that can be administered to mice in a suitable volume. This dose showed ~100% lysis in the CTL study (Fig 4).

Comment 5:

How did DODAP selectively accumulate the mRNA in the spleen? What is the mechanism behind it?

Based on our data, the lipid composition of DODAP/DOPE-LNPs affects the efficiency of gene expression and immune stimulation rather than the biodistribution. It seems that there are two possible factors to explain this phenomenon; one is about intra-tissue distribution, and the other is about transfection efficiency in different cell types. The total amount of gene expression in a tissue can change if the cells taking up LNPs change because there is a cell-line dependent difference in transfection efficiency. The optimal composition of lipid-based carriers is different depending on the cell type. In that present case, DODAP/DOPE-LNP is thought to be more efficient in transfecting antigen-presenting cells in the spleen than in other cell types.